EMBO
Molecular Medicine

# *In vivo* generation of a mature and functional artificial skeletal muscle

Claudia Fuoco[1], Roberto Rizzi[2,3], Antonella Biondo[1], Emanuela Longa[4], Anna Mascaro[4], Keren Shapira-Schweitzer[6], Olga Kossovar[6], Sara Benedetti[5,‡], Maria L Salvatori[1], Sabrina Santoleri[1,†], Stefano Testa[1], Sergio Bernardini[1], Roberto Bottinelli[4,7], Claudia Bearzi[2,3], Stefano M Cannata[1,***], Dror Seliktar[6], Giulio Cossu[5,8,**] & Cesare Gargioli[1,2,*]

## Abstract

Extensive loss of skeletal muscle tissue results in mutilations and severe loss of function. *In vitro*-generated artificial muscles undergo necrosis when transplanted *in vivo* before host angiogenesis may provide oxygen for fibre survival. Here, we report a novel strategy based upon the use of mouse or human mesoangioblasts encapsulated inside PEG-fibrinogen hydrogel. Once engineered to express placental-derived growth factor, mesoangioblasts attract host vessels and nerves, contributing to *in vivo* survival and maturation of newly formed myofibres. When the graft was implanted underneath the skin on the surface of the *tibialis anterior*, mature and aligned myofibres formed within several weeks as a complete and functional extra muscle. Moreover, replacing the ablated *tibialis anterior* with PEG-fibrinogen-embedded mesoangioblasts also resulted in an artificial muscle very similar to a normal *tibialis anterior*. This strategy opens the possibility for patient-specific muscle creation for a large number of pathological conditions involving muscle tissue wasting.

**Keywords** artificial skeletal muscle; mesoangioblasts; PEG-fibrinogen
**Subject Categories** Musculoskeletal System; Regenerative Medicine

## Introduction

Tissue engineering aims to create a microenvironment similar to the one where organogenesis took place. This requires the combined use of progenitor cells and biomaterials that allow optimal cell-matrix interactions, thus promoting better engraftment, differentiation and, consequently, better tissue generation. Skin, bone and cartilage, and recently trachea and urinary bladder, are successful examples of this strategy (Horch *et al*, 2000; Koop *et al*, 2004; Vangsness *et al*, 2004; Bader & Macchiarini, 2010; Watanabe *et al*, 2011). In contrast, tissue engineering for skeletal muscle still represents a difficult task despite its potential for the treatment of a variety of pathological conditions including post-traumatic muscle damages, post-surgery tissue ablation and incontinent sphincters (Guettier-Sigrist *et al*, 1998; Rizzi *et al*, 2012). Three-dimensional skeletal muscle tissue has been developed exploiting the influence of mechanical stretch (Vandenburgh & Kaufman, 1979), revealing the pivotal role played by kinetic forces on myofibre organization and muscle maturation. For example, Powell and colleagues (Powell *et al*, 2002) improved engineered three-dimensional (3D) human skeletal muscle on a collagen and matrigel scaffold by applying mechanical stimulation, thus increasing elasticity, and cross-sectional area (CSA). Moreover, several other recent developments, using decellularized natural scaffold to repair massive muscle injury, exploit host stem cells regenerative capabilities. The latter approach has shown only partial integration with damaged muscle and limited vascularization and innervation at the interface between the artificial and the host muscle (Corona *et al*, 2012; Sicari *et al*, 2012). Hence, this method still needs optimization, especially in terms of supporting blood vessel and nerves for artificial tissue survival and function. To overcome this hurdle, we looked for an innovative strategy exploiting a biomaterial able to promote myogenic cell differentiation *in vivo* so that angiogenesis and innervation may occur during muscle fibre formation and maturation.

1 Department of Biology, Tor Vergata Rome University, Rome, Italy
2 IRCCS MultiMedica, Milan, Italy
3 Cell Biology and Neurobiology Institute, National Research Council of Italy, Rome, Italy
4 Department of Molecular Medicine and Interdepartmental Centre for Research in Sport Biology and Medicine, University of Pavia, Pavia, Italy
5 Department of Cell and Developmental Biology, University College London, London, UK
6 Faculty of Biomedical Engineering, Technion—Israel Institute of Technology, Haifa, Israel
7 Fondazione Salvatore Maugeri (IRCCS), Scientific Institute of Pavia, Pavia, Italy
8 Institute of Inflammation and Repair, University of Manchester, Manchester, UK
 *Corresponding author. Tel: +39 6 72594815; E-mail: cesare.gargioli@uniroma2.it
 **Corresponding author. Tel: +44 161 3062526; E-mail: giulio.cossu@manchester.ac.uk
 ***Corresponding author. Tel: +39 6 72594815; E-mail: cannata@uniroma2.it
 †Present address: Biocenter Oulu, Institute of Biomedicine, University of Oulu Finland
 ‡Correction added on 10 March 2015, after first online publication: author affiliations have been corrected.

# Results

### PEG-fibrinogen enhances Mabs myogenic differentiation

A photopolymerizable hydrogel based upon polyethyleneglycol (PEG) and fibrinogen (PEG-fibrinogen: PF) (Almany & Seliktar, 2005; Fuoco *et al*, 2012; Seliktar, 2012) was combined with vessel-associated muscle progenitors, termed mesoangioblasts (Mabs), which are able to undergo robust myogenesis *in vivo* and *in vitro*. PF, which is clinically approved, creates a favourable microenvironment by coupling natural and synthetic features, adjustable to the needs of each specific cell type; Mabs are also already clinically approved, and a clinical trial (Eudract Number: 2011-000176-33) for Duchenne Muscular Dystrophy has been recently completed (G. Cossu *et al*, in preparation). The adult mouse C57 Mabs (Díaz-Manera *et al*, 2010) embedded into PF showed a remarkable muscle differentiation (Fig 1A and B) and were therefore selected for subsequent experiments and indicated as mouse Mabs (mMabs). A cylindrical silicon mould 0.5 cm high × 0.2 cm diameter was loaded with 100 μl solution of 8 mg/ml PF with $5 \times 10^5$ mMabs transduced with a lentiviral vector encoding the reporter gene nuclear β-galactosidase (mMabs-nLacZ); the constructs were formed by exposing the moulds to non-toxic long-wave UV light, as described in the Materials and Methods. The constructs were cultured for 24 h in serum-supplemented growth medium and then transferred into serum-depleted differentiation medium for 5 days in order to promote muscle fibre formation (Fig 1A and B). The Mabs rapidly underwent cell fusion in the PF within 24 h as documented by the time course performed in parallel with mMabs cultured into 2D standard plastic culture (TCP) versus mMabs encapsulated into PF (Supplementary Fig S1). Mabs moved well inside the gel (Supplementary Movie S1) and showed a much faster differentiation in the 3D PF cultures (Supplementary Fig S1D–F) than TCP one (Supplementary Fig S1A–C), leading to the formation of mature well-differentiated contractile myotubes. Of notice, even newly formed myotubes contracted spontaneously (Supplementary Movie S2) something usually not observed on TCP. We did not quantify proliferation and differentiation inside PF, as compared to control cells grown on plastic, due to the difficulty counting cells in the multiple focal plans within the gel. However, after 5 days *in vitro*, immunofluorescence analysis for myosin heavy chain (MyHC) and Western blot analysis revealed robust expression of skeletal muscle-specific proteins such as tropomyosin, troponin, creatine kinase and MyHC (Fig 1C and D). Furthermore, after 15 days of 3D culture into 8 mg/ml PF, mMabs generated a very thick three-dimensional network of fibres able to contract the plug as whole (Supplementary Movie S3).

### PlGF promotes *in vivo* vessel recruitment

Undifferentiated mMabs embedded in PF were subsequently implanted under the back skin of 2-month-old RAG2/γchain null mice (Cao *et al*, 1995). Immunodeficient mice were used to prevent immune reaction against the bacterial LacZ gene used as reporter. Angiogenesis in the implanted tissue was enhanced by transduction of the mMabs-nLacZ with a lentiviral vector expressing placental-derived growth factor (PlGF) that we had previously shown to enhance angiogenesis of transplanted mMabs (Gargioli *et al*, 2008). To rule out the possibility that PlGF expression may interfere with mMabs differentiation, we performed immunofluorescence analysis for MyHC which showed similar myogenic differentiation of both the wild-type mMabs and mMabs-nLacZ expressing PlGF (mMabs-nLacZ/PlGF) (Supplementary Fig S2A and B). *In vivo* analysis of dorsal subcutaneous PF implants loaded with $1.5 \times 10^6$ mMabs-nLacZ/PlGF revealed increased blood vessel density in comparison with control mMabs-nLacZ (Supplementary Fig S2C–G), quantified by counting the number of blood vessels, stained for VE-cadherin per muscle fibre (Supplementary Fig S2H). Robust muscle differentiation of cells embedded into PF was observed, with enhanced differentiation in mMabs-nLacZ/PlGF comparing with unmodified mMabs (Supplementary Fig S2I–L). The mMabs-nLacZ/PlGF showed an increased number of MyHC-positive myofibres in the centre of the implant due to an enhanced vascularization and thus to improved oxygenation and nutrition, while unmodified mMabs generated fewer and smaller muscle fibres in the centre of the implant (Supplementary Fig S2I and J). Because of these results, only mMabs expressing PlGF were utilized for subsequent *in vivo* studies. Hydrogel resorption was analysed in subcutaneous implants of $1.5 \times 10^6$ mMabs-nLacZ/PlGF encapsulated into 50 μl of 8 mg/ml PF: after 3 days, the PF regular structure still surrounded the cells, whereas after 7 days, the PF was almost completely resorbed (Supplementary Fig S2M and N). In both conditions (with or without PlGF), newly formed myofibre was randomly oriented and this did not result in any structure anatomically recognizable as a skeletal muscle (Supplementary Fig S2K and L).

### Contracting muscle surface as 'anatomical bioreactor' promoting fibres alignment

We reasoned that a functional, contractile muscle could influence the implant evoking myofibres alignment in response to its contractile activity mimicking a stretching stimulus. Thus, we implanted mMabs-nLacZ/PlGF ($1.5 \times 10^6$) in PF constructs (cylindrically shaped, 0.5 cm high × 0.2 cm diameter) under the skin covering the surface of the *tibialis anterior* (TA) to exploit its contractile activity and then promoting artificial muscle fibres alignment recapitulating the normal skeletal muscle tissue architecture (Supplementary Fig S3 and Fig 2A). The resulting structures were collected at early (4 weeks) and late stages (8 weeks) after implantation. The implants were completely incorporated by the host TA epimysium (Fig 2B) and were revealed only when tendons were cut (Fig 2C). The resulting structure showed size and morphology very similar to the underlying TA muscle (Fig 2C and D). Macroscopic observation of the whole muscle showed a network of blood vessels on the muscle-like tissue (Fig 2E), while systemic ink injection via recipient mouse femoral artery confirmed the expected connection with the host vascular tree (Fig 2F). Western blot analysis of muscle and vessel protein expression was performed on crude extracts from 8-week implanted tissue and revealed an expression pattern comparable to control host TA (Fig 2G and H). The development of a mature and vascularized artificial muscle tissue was further confirmed by real-time quantitative PCR, which revealed comparable expression levels of all the muscle-specific genes examined (Supplementary Fig S4). β-Galactosidase staining on histological sections of early (4 weeks) and mature (8 weeks) artificial muscle revealed LacZ-positive donor nuclei centrally located at early stages but peripherally located at 8 weeks, whereas cells derived from the

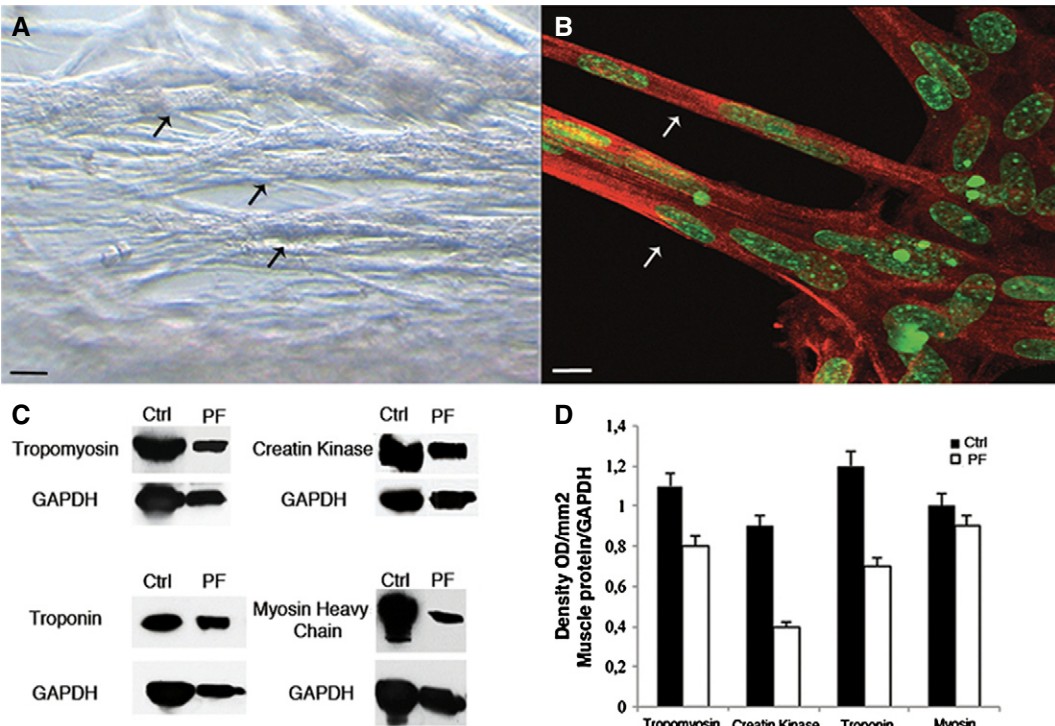

**Figure 1.** *In vitro* characterization of Mabs grown inside PF hydrogels.

A    Phase contrast images of Mabs cultured into PF hydrogels (5 days): cells formed a thick three-dimensional myotube network with numerous thick myofibres (arrows). Scale bar: 50 μm.

B    Confocal immunofluorescence analysis with an antibody against myosin heavy chain (MyHC) (red) and SYTOX green which labels nuclei (green), showing multi-nucleated, mature muscle fibres (arrows) developed from Mabs in PF (5 days). Scale bar: 50 μm.

C, D    Western blot and densitometry analysis of different protein extracts of Mabs differentiated inside PF constructs (PF) showing expression levels comparable to adult TA (Ctrl). Group of *n* = 6 of Mabs differentiated inside PF and host TA has been tested, densitometric analysis from *n* = 3 different Western blots gauging the level of expression of muscle markers is presented as means ± standard error.

Source data are available online for this figure.

host were located mainly in the interstitial structures where the LacZ-positive cells were mostly absent (Fig 2I–L). Immunofluorescence analysis with anti-laminin antibodies revealed an incomplete basal lamina and residual interstitial tissue at 4 weeks; in contrast, at 8 weeks, the artificial muscle showed an almost normal muscle tissue organization with well-patterned basal lamina (Fig 2M–P). Immunostaining for neurofilaments and bungarotoxin staining (which binds the acetylcholine receptor) (Supplementary Fig S5A) were performed at early and late stages and showed a progressive maturation of the developing synapses; only axons were detected at early stages (Supplementary Fig S5B), whereas bungarotoxin-positive, neuromuscular plaques were detected at late stages (Supplementary Fig S5C). Neuromuscular synapses were also labelled by immunofluorescence and visualized by confocal microscopy; the isosurfaces showed the formation of mature neuromuscular plaques in the generated artificial new muscle (Fig 2Q). Moreover, the artificial muscle contained many small vessels identified by immunostaining with antibodies against smooth muscle actin (SMA) and vascular endothelial cadherin (VE-Cad) (Fig 2R). Despite an apparently normal muscular morphology and gene expression pattern (Supplementary Fig S4), careful examination of the artificial muscle and histological longitudinal sections did not reveal tendons at the extremities (Fig 2S). Interestingly, several nLacZ donor nuclei

were detected in the underlying TA—which was not injured during the implantation—confirming that myogenic cells may colonize adjacent muscles in small rodents (Hughes & Blau, 1990). This finding may also explain why only a part of the artificial muscle had LacZ-positive nuclei, due to incomplete transduction (about 80%) of donor mesoangioblasts and possible cell migration from the underlying TA (Supplementary Fig S6A–F). Moreover, the new muscle tissue showed re-constituted Pax7-positive satellite cell pool, in part expressing β-galactosidase (Supplementary Fig S6G–I). Therefore, in order to verify host contribution to artificial tissue generation, we performed experiments into ubiquitous GFP mice background using PF-embedded mMabs-PlGF. Because of an immune response against implanted PF and mMabs-derived structure, we could analyse only short-time engraftment (15 days), when we observed myogenic differentiation of implanted mMabs in the absence of GFP-positive cells that could be detected in the adjacent tissue (Supplementary Fig S7). Furthermore, the artificial muscle ability to regenerate was investigated by cardiotoxin injection that caused myofibre degeneration, followed by muscle fibre regeneration with kinetics comparable to that of a normal TA (Supplementary Fig S8). After damage, the artificial muscle and the underlying TA contain satellite cells with similar myogenic potency: magnetically sorted satellite cells showed LacZ expression when isolated from the artificial

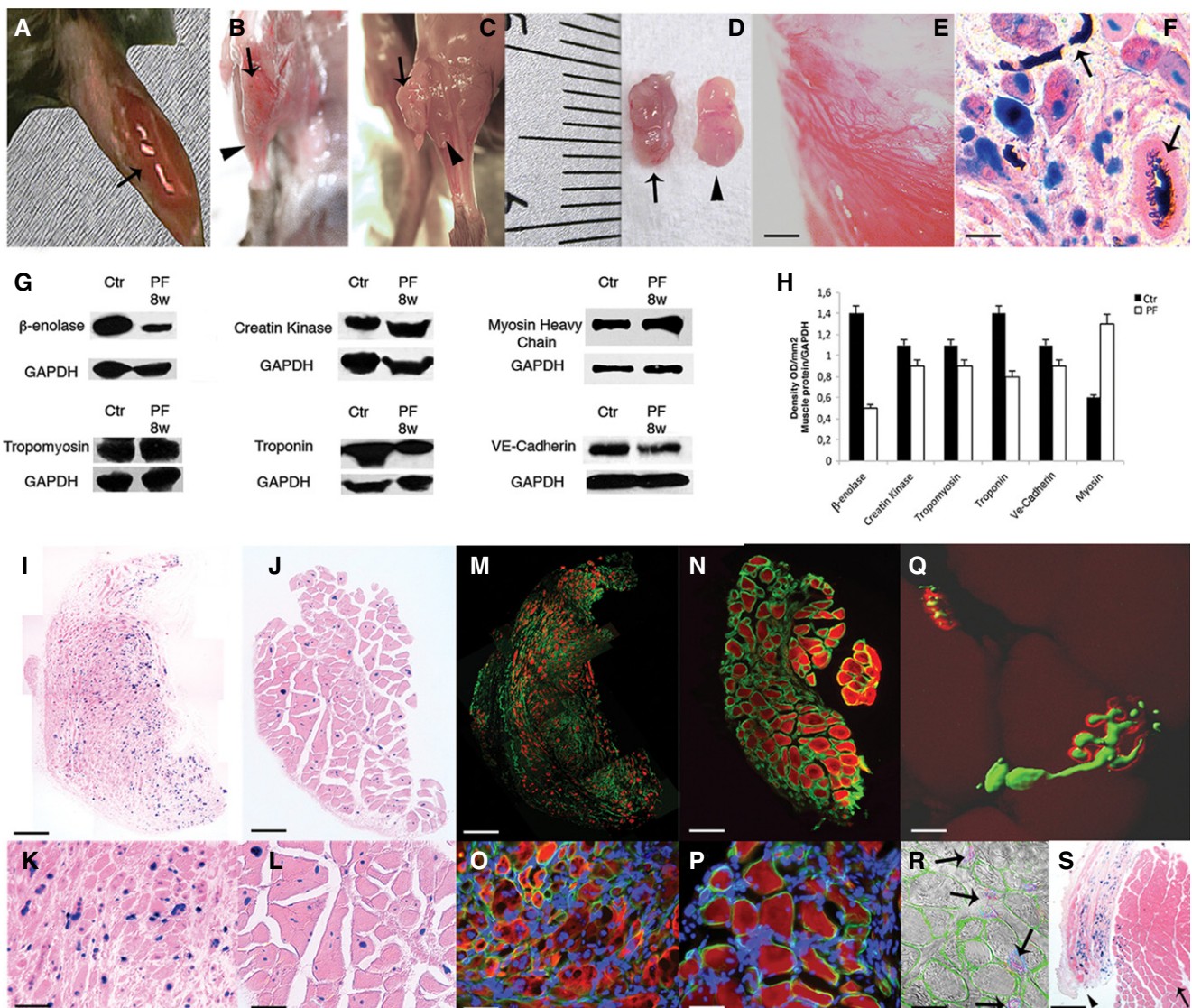

**Figure 2. Implanted constructs of Mabs in PF on the surface of TA.**

A Gross morphology of the PF implant containing Mabs-nLacZ/PlGF immediately after implantation on the surface of underlying TA (arrow).

B, C Artificial muscles (arrows) developed within 8 weeks over the host TA surface (arrowheads), encapsulated by the host perimysium (B) and released after tendon resection of the TA (C).

D The isolated artificial muscle from the TA surface (arrow) is comparable in size to the underlying host TA (arrowhead).

E Stereomicroscopy images revealing blood vessel formation in the artificial muscle. Scale bar: 100 μm.

F X-Gal and H&E staining on artificial muscle section revealing vessel connection with host vasculature by Indian ink, injected into the femoral artery immediately before sacrifice. Scale bar: 20 μm.

G Western blot analysis of protein extracts from artificial muscles (PF) and host TA (Ctrl) reveals remarkable expression of muscle- and vessel-specific proteins (*n* = 5).

H Densitometric analysis from *n* = 3 different Western blots gauging the level of expression of muscle- and vessel-specific proteins; presented values are expressed as means ± standard error.

I–L X-Gal and H&E staining of artificial muscle sections at 4 (I, K) and 8 weeks (J, L) after implantation, showing donor origin of muscle cells (nLacZ+ nuclei) and correlation between implant duration and myofibre maturation (I, J images are the results of the collage of several photos). Scale bar: (I, J) 200 μm, (K, L) 100 μm.

M–P Immunofluorescence analysis on adjacent sections revealing laminin organization (green) and myosin heavy chain (MyHC) expression (red) at 4 weeks (M) and 8 weeks (N) after grafting (M, N displayed items are obtained from a multi-photo collage); DAPI staining (blue) shows centrally located nuclei in developing fibres (O) and peripheral nuclei in mature myofibres (P). Scale bar: (M, N) 200 μm, (O, P) 50 μm.

Q Confocal isosurface image of neurofilament (green) and bungarotoxin (red) immunofluorescence in the artificial muscle section shows a mature neuromuscular plaque. Scale bar: 10 μm.

R Immunofluorescence against smooth muscle actin (blue), laminin (green) and VE-Cad (red) superimposed on phase contrast image shows blood vessels (arrows) adjacent to the myofibres in the artificial muscle section. Scale bar: 20 μm.

S X-Gal staining on mature artificial muscle (arrowhead) and the underneath host TA (arrow) revealing no tendon development at the extremity of the artificial tissue; implanted LacZ-positive Mabs are present within the host TA. Scale bar: 200 μm.

Source data are available online for this figure.

muscle but not from the host TA and underwent myogenesis with comparable efficiency (Supplementary Fig S9 and Supplementary Table S1). Finally, in order to understand whether artificial muscle would respond to hypertrophic or atrophic stimuli like a normal muscle, clenbuterol administration and denervation were performed (Supplementary Fig S10). The analysis revealed a striking atrophic response to sciatic nerve resection both in the artificial muscle and in the underlying TA (Supplementary Fig S10G–J), while the clenbuterol treatment showed only a modest, which did not reach statistical significance in both artificial and underlying muscles, possibly due to inadequate response time or concentration (Supplementary Fig S10C–F); quantification obtained evaluating CSA average of the treated artificial and TA muscles in ten non-adjacent sections (Supplementary Fig S10A and B).

## Artificial muscle closely resembles a natural skeletal muscle

Ultrastructural and functional analyses were conducted to evaluate to what extent the artificial muscle resembles a normal adult muscle. Electron microscopy analysis revealed a completed sarcomerogenesis typical of mature skeletal muscle while staining with blu-O-gal revealed the presence of donor-derived LacZ-positive nuclei adjacent to transversely sectioned sarcomeres (Fig 3A and B). Single artificial myofibres were isolated from bunches of artificial muscle fibres (LacZ-positive) and showed normal cross-striation (Fig 3C and D). A subset of the artificial muscle fibres ($n = 20$) was stained by immunofluorescence with antibodies against β-galactosidase to assess the origin of their nuclei: results showed that the large majority of nuclei were of donor origin (based on nuclear LacZ expression), but few unlabelled nuclei, whose origin remain unclear, were also present (Fig 3E and F) (Hughes & Blau, 1990). A large number of nLacZ-positive artificial muscle fibres ($n = 153$) and underlying TA muscle control fibres ($n = 103$) from 4-month-old C57 RAG2/γchain$^{-/-}$ mice were mounted in a set-up which enabled viewing their striation pattern at 320-times magnification, determining CSA, specific force (Po/CSA) and maximum shortening velocity (Vo) (Bottinelli *et al*, 1996; Pellegrino *et al*, 2003). The striation pattern, CSA, Po/CSA and Vo of muscle fibres from artificial muscle and from control muscle were indistinguishable (Fig 3G–I). Po/CSA values were similar in single fibres from the artificial muscle and the underlying control muscle. Moreover, the values of specific force reported in Fig 3G are fully consistent with our previous findings on muscle fibres from normal TA muscles of mice (Torrente *et al*, 2004). Likewise, the value of Vo, which is mainly dependent on the MyHC isoform content, indicated that the kinetic of actomyosin interaction was virtually identical to that of the normal adult fibres of the underlying TA (Bottinelli *et al*, 1994; Pellegrino *et al*, 2003; Torrente *et al*, 2004). At the end of the functional analysis, the MyHC isoform content of each fibre was also determined by SDS–PAGE. All fibres analysed were found to contain adult MyHC isoforms and almost exclusively MyHC-2B, which is the most expressed MyHC isoform in fast mouse muscles (Pellegrino *et al*, 2003) (Fig 3J and K).

## Human Mabs-derived artificial muscle

Human Mabs (hMabs) were isolated as described in the material and methods (Tonlorenzi *et al*, 2007) and combined with PF as

described for mouse Mabs. Short-term analysis showed that the human cells behave like their murine counterparts. Specifically, myogenic differentiation of the hMabs in PF was observed using *in vitro* cultures in as few as 5 days (Supplementary Fig S11A–F). Moreover, hMabs exhibited an excellent capacity to form a new human-derived artificial muscle *in vivo* when implanted into host recipient immunodeficient SCID mice (Supplementary Fig S11G–I). However, even in an immunodeficient background, over time the xenogeneic graft with time attracted murine macrophages and other non-lymphoid cells that infiltrated and prevented myogenic maturation of the new muscle. Despite this problem, specifically related to xenotransplantation, these data show that this method can be applied to human cells, setting the stage for future clinical application, even though extension of this method to large human muscles may be technically very demanding and may require significant additional work. Overall, the data presented above establish a new paradigm for skeletal muscle tissue engineering, by showing that it is possible and relatively simple to create an artificial muscle by exploiting a contracting muscle surface as an 'anatomical bioreactor'. In approximately 2 months, the artificial muscle matures to become morphologically and functionally very similar to the underlying, supporting host muscle. However, lack of tendons prevented any functional test on the artificial muscle *in vivo* or *ex vivo*.

## Artificial muscle replacing the endogenous TA promotes functional recovery

Hence, in order to test whether the combination of mesoangioblasts and PEG-fibrinogen may lead to a biomechanically functioning artificial muscle, we performed implantations into a severe injury muscle model. The TA of 2-month-old SCID mice was almost entirely ablated surgically [80–90% of the TA was removed, dislodged tissue weighting an average of 85 mg (TA weight ~100 mg) measured in five ablated mice, Supplementary Fig S12], leaving the tendons intact in place to anchor the implant. The TA was replaced with an implant comprised of $3 \times 10^6$ mMabs expressing PlGF (mMabs-PlGF) encapsulated into 8 mg/ml PF within a total volume of 50 μl (Supplementary Movie S4 and Supplementary Fig S13). Acellular PF gels were used as controls in order to analyse the ability of the PF scaffold to induce host cell migration and artificial muscle formation. The treated mice were monitored for 6 months for their motor activity, running distance and limb strength. Histological analysis was also performed at early and late time points after the TA replacement (Fig 4). Soon after the intervention, all mice regain the ability of walking, but they could not flex the foot as expected after the resection of the TA preventing normal ambulatory activity (Supplementary Movie S5). Six months after TA removal, mice treated with a-cellular PF showed a complete lack of regeneration (Fig 4A), while the mice treated with PF-embedded mMabs-PlGF showed the presence of a new artificial muscle of approximately the same size as the ablated TA (Fig 4B and C). Immunofluorescence and histological studies were performed at early (10 days) and late (6 months) time points after TA removal (Fig 4D–O and S–Z) in order to verify TA ablation and its regeneration onset at very early time point, and the artificial TA status at the end of reconstruction process. The results obtained were comparable to those observed with the

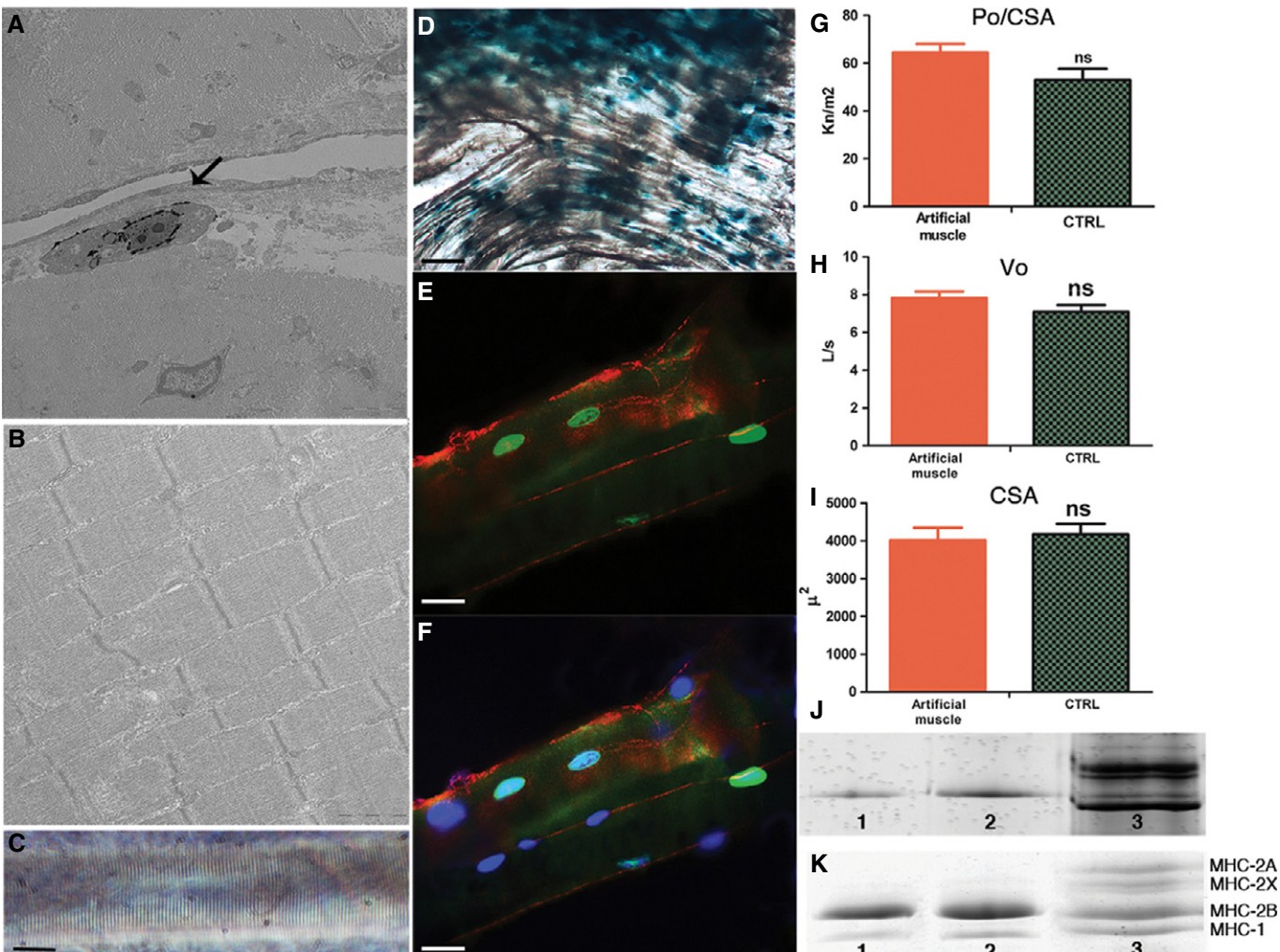

**Figure 3.  Structural and functional analysis of artificial muscle and underlying TA control muscle.**

A    Electron microscopy with Bluo-Gal staining reveals a LacZ-positive donor nucleus (arrow) peripheral to transversely sectioned sarcomeres of the artificial muscle. Scale bar: 1 μm.

B    Longitudinal EM section of an artificial muscle showing mature sarcomere myofibril organization. Scale bar: 0.5 μm.

C    A single fibre from an artificial muscle showing a striation pattern typical of a mature myofibre. Scale bar: 10 μm.

D    X-Gal staining of freshly dissected artificial muscle bundle at 8 weeks showing abundant LacZ-positive donor nuclei. Scale bar: 100 μm.

E    Immunofluorescence analysis of LacZ-positive nuclei (green) and dystrophin (red) in an individual muscle fibre after functional analysis. Scale bar: 10 μm.

F    The same fibre shown in (E) is counterstained with DAPI to show that the interstitial nuclei (blue) are of donor origin. Scale bar: 10 μm.

G–I    Specific force (Po/CSA) (G), maximum shortening velocity (Vo) (H), and cross-sectional area (CSA) (I) of single fibres ($n$ = 153) from artificial muscles and control (CTRL) muscles ($n$ = 103).

J    Silver-stained SDS–PAGE (8%) gel revealing myosin heavy chain (MyHC)-2B isoform expression in two single fibres of artificial muscles (lanes 1, 2) and a bulk muscle control sample from soleus muscle containing all four adult MyHC isoforms indicated in (K) (lane 3).

K    Comassie blue-stained SDS–PAGE (8%) of bulk samples of two artificial muscles (lanes 1, 2) and of a bulk control muscle sample (lane 3), as in (J), showing different MyHC isoform expression.

Data information: In (G–I), values are expressed as means ± standard error, significance was tested using Student's $t$-test, and $P$ > 0.05 was considered not significant (ns).

supernumerary muscle grown on the surface of the TA (Supplementary Fig S14) despite a different experimental approach. The new artificial muscle also revealed a time-dependent maturation with few MyHC-positive myofibres at 10 days after TA removal (Fig 4G–I). Number and size of MyHC$^+$ fibres increased with time and reached a complete maturation after 6 months, being also surrounded by mature basal lamina but still with few centre-nucleated myofibres (Fig 4M–O). The mice treated with a-cellular

PF gels as control showed an almost complete absence of muscle, showing only very few fibres both at early and at late time points (Fig 4D–F and J–L). Histological analysis revealed a rapid formation of small centre-nucleated myofibres in mice treated with PF + mMabs, already visible at 10 days after TA ablation (Fig 4U and V). Control mice did not show any regenerating muscle fibres, but a few remaining mature fibres were observed residual after massive TA removal (Fig 4D, E, S and T). At the later stages, the

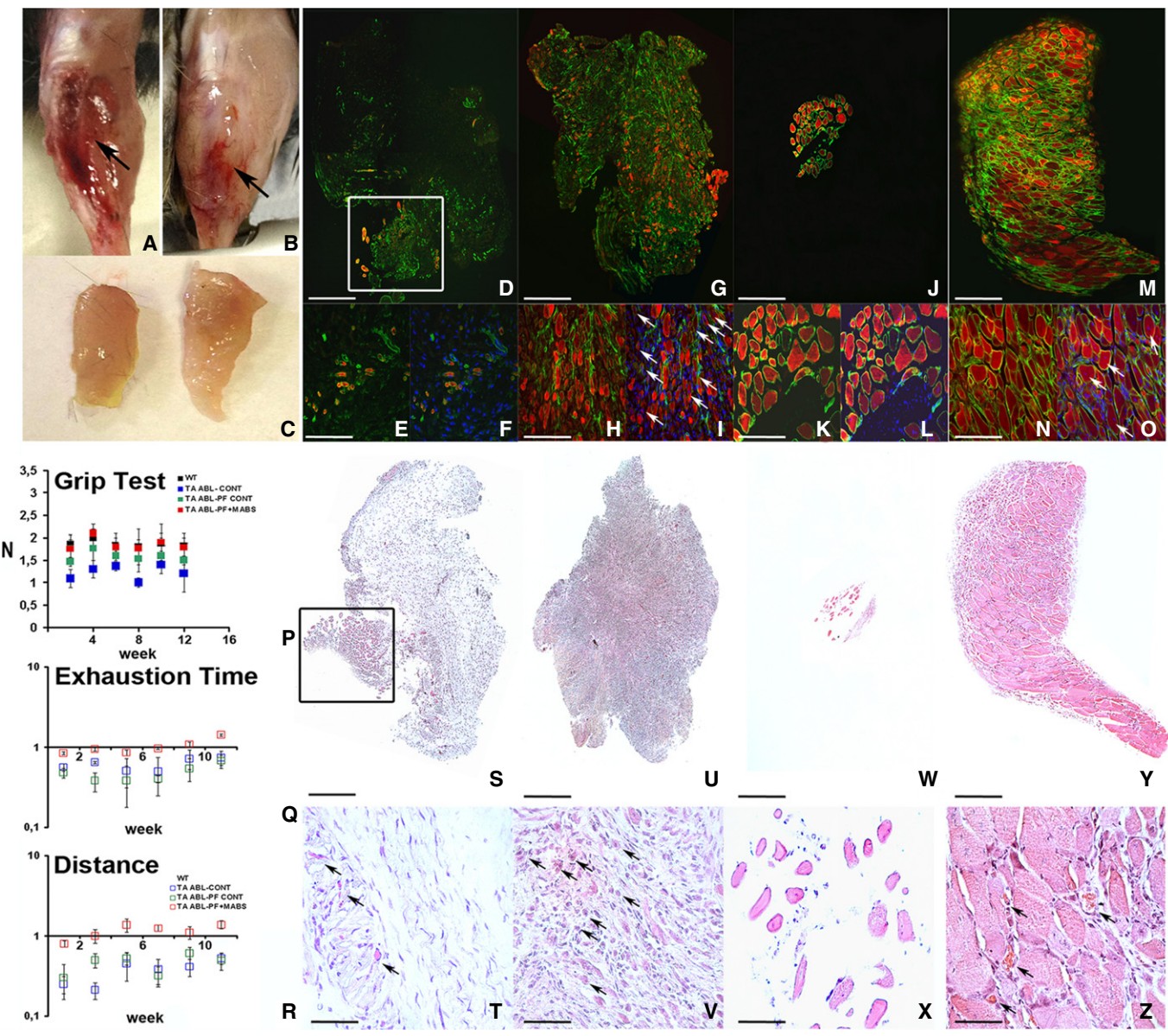

**Figure 4.**

reconstituted artificial muscle showed an almost normal morphology (Fig 4Y and Z) with peripherally nucleated fibres, surrounded by healthy blood vessel network (arrows Fig 4Z) and a well-formed basal lamina. Functionality of the artificial muscle was tested by grip and tread mill tests performed for twelve weeks which revealed the strength and resistance to running of the PF + mMabs-treated mice almost identical to wild-type healthy mice (Fig 4P–R). It is interesting to note that, even in the absence of an underlying contracting muscle, the artificial muscle formed with aligned and thus functional myofibres, possibly because of the residual contractile activity of the neighbouring muscles.

## Discussion

This work represents a significant step towards the generation of an artificial but fully functional, vascularized and innervated skeletal muscle. Previous and more recent attempts demonstrated the possibility to obtain an artificial muscle-like tissue vascularized and partially functional, although innervation analysis was not investigated (Levenberg *et al*, 2005; Carosio *et al*, 2013). Moreover, the artificial muscle architecture shown in these previous reports is clearly different from the natural muscle morphology and structural organization. In contrast, our strategy results in a gross morphology and histological and ultra-structural architecture, which are extremely similar if not indistinguishable from a natural skeletal muscle. Moreover, our work shows the unprecedented possibility to repair a massive muscle ablation with a functional '*restitutio ad integrum*'. The fact that the satellite cell compartment was mainly replenished by LacZ-positive host-derived cells supports the promising efficacy of the reported technology and favours artificial muscle tissue integration and functionality *in vivo*. However, it should be noted that such promising efficacy has been so far demonstrated for a small rodent muscle only. Furthermore, by implanting a-cellular PF, we

**Figure 4.  PF-embedded Mabs grafted in an ablated TA injury, showing full recovery of muscular morphology and functionality.**

A, B    Gross morphology of the TA injury at 6 months after massive muscle ablation, revealing the new artificial TA regeneration (arrow) when grafted with PF-embedded Mabs-PlGF (B) and absence of TA muscle (arrow) when treated with empty PF as a control (A).

C    PF-embedded Mabs-PlGF-derived artificial TA (right) showing comparable size to the wild-type normal TA (left).

D–O    Immunofluorescence against myosin heavy chain (MyHC) (red) and laminin (green) on mice TA cross sections at early (10 days) (D–I) and later stage (6 months) (J–O) after massive removal of the muscle (D, G, J, M are the results of a collage of several photos). TA regeneration at early stages showed very few residual myofibres after implantation of acellular PF in place of the ablated TA (white box in D), an enlarged view (E), and DAPI-counterstained view (F) reveals the infiltration of undifferentiated mono-nucleated cells. Conversely, a PF-embedded Mabs-PlGF graft (G) demonstrated early-stage regeneration with several MyHC-positive (red) regenerating myofibres surrounded by laminin (green) (H), most of these (arrows) being centre-nucleated (DAPI labelled in I). At the later stages, acellular PF implanted mice (J) showed a very small portion of muscle tissue likely derived from residual myofibres from TA removal, higher magnification (K) revealed very few mature myofibres with peripheral nuclei (DAPI labelled in L). Contrary, the PF-embedded Mabs-PlGF treatment (M) showed a complete replacement of the ablated TA with the recovered native structure of MyHC-positive myofibres surrounded by laminin (N), some of these (arrows) still being centre nucleate (DAPI labelled in O). Scale bar: (D, G, J, M) 500 μm, (E, F, H, I, K, L, N, O) 100 μm.

P–R    Functional analysis demonstrated the effective ability of the PF and Mabs-PlGF combination to generate a complete and functional artificial TA replacing the ablated TA injury. (P) The fore-hind limb grip test diagram reveals functional grab force recovery for mice treated with PF-embedded Mabs-PlGF (TA ABL-PF+MABS) after only 4 weeks post-TA injury, being comparable with not-ablated wild-type (WT) mice and significantly higher than control ablated mice (TA ABL-CONT) or mice implanted with void PF (TA ABL-PF CONT) ($n = 10$ mice per group). (Q, R) The treadmill analysis of the treated mice relative to wild-type not-ablated (wt) shows both exhaustion time and distance indicative of a full recovery of motor functionality already at 3 weeks after TA ablation with an increase in time and distance run for mice treated with PF-embedded Mabs-PlGF, while control mice (TA ABL-CONT or TA ABL-PF CONT) showed a much lower resistance ($n = 10$ mice per group). The values shown in the diagrams are expressed as means $\pm$ standard error and statistical significance was tested using Student's *t*-test ($P < 0.05$ was considered significant).

S–Z    Morphological analysis demonstrated the effective ability of the PF and Mabs-PlGF combination to generate a complete and functional artificial TA replacing the ablated TA injury. H&E staining of TA injury in mice by cross section at an early (10 days) (S–V) and late (6 months) (W–Z) time point after ablation (in S, U, W, Y displayed items are obtained from a multi-photo collage). At the early time point, the acellular PF-treated mice revealed few mature muscle fibres that survive the massive ablation (box in S) signs of continued muscle degeneration (arrows in T), whereas PF-embedded Mabs implanted mice showed remarkable muscle regeneration indicated by centre-nucleated myofibres (arrows in V). At later stages, the control mice grafted with acellular PF showed an almost devoid TA (W) with very few muscle fibres (X), in contrast to PF-embedded Mabs, which exhibited a complete muscle replacement generating an artificial TA (Y), enlarged view (Z) proved the complete and mature organization of the newly formed muscle showing capillary and blood vessel around myofibres (arrows). Scale bar: (S, U, W, Y) 500 μm, (T, V, X, Z) 50 μm.

Source data are available online for this figure.

were able to demonstrate that the scaffold matrix by itself is not able to support formation of new muscle mediated by host stem cell migration, thus showing that the combination of PF and Mabs only can generate a complete and functional artificial muscle. Even if in Mabs + PF-derived artificial muscle, there is a host cell co-participation, this is mainly in the interstitial compartment and probably advantageous for clinical reconstructive strategy. This by no means implies that other biomaterials and other myogenic cell types may not result in a similar functioning muscle, though this remains to be tested. In the case of human cells, inflammatory host cell infiltration prevented complete maturation of the human artificial muscle, but we believe this to be a problem of xenotransplantation and not relevant to an intraspecific, possibly autologous context.

Although differences were detected with *bona fide* mature muscles such as the absence of MyHCs 2A and increased density of nerves, both consistent with a still partially immature phenotype, the contractile properties of newly formed fibres were indistinguishable from those of the underlying and wild-type TA. Clearly, a number of issues still remain to be solved before this artificial muscle may be used in a clinical setting. Importantly, tendons should be developed *in vivo* by integrating the edge of the muscle that already contains connective tissue to the existing tendons and eventually adding tendon fibroblasts at the edge. Equally important is the use of larger animal models that should be developed for testing efficacy of this approach before clinical translation. Finally, current Good Manufacturing Practices (cGMP) will have to be developed for this combination of cells and biomaterial, even though both mesoangioblasts and PEG-fibrinogen are in clinical experimentation. The discussed strategy offers the possibility of creating *de novo* a functional artificial muscle in those anatomical districts where it is needed. This opens up new scenarios for regenerative medicine: ablated or irreversibly damaged muscles could be replaced with patient's own cells and biomaterial could be implanted on top of a residual muscle, adjacent to the damaged area where the new muscle will be later transferred with nerves and vessels maintained. Clinical translation of this procedure should be possible in a relatively near future, as already shown by Sicari and colleagues (Sicari *et al*, 2014) experimenting in human patient material scaffold to promote muscle reconstruction and function restoration in a massive muscle ablation context.

## Materials and Methods

### Cells and culture conditions

We cultured mouse mesoangioblasts (Mabs) on Falcon dishes at 37°C with 5% $CO_2$ in DMEM GlutaMAX (Gibco) supplemented with heat-inactivated 10% foetal bovine serum (FBS), 100 international units/ml penicillin and 100 mg/ml streptomycin. We transduced the cells with third-generation lentiviral vectors encoding β-galactosidase (nLacZ) and/or PlGF as described (Gargioli *et al*, 2008). Human Mabs (hMabs) were isolated and cultured as described before (Tonlorenzi *et al*, 2007; Tedesco *et al*, 2012); briefly, hMabs were isolated from cultured muscle biopsies provided by Dr. Roberto Biagini head of orthopaedic oncology Rome IRE-ISG after obtaining patient informed consensus. The human samples study protocol was evaluated by the ethical committee of the Rome IRES-ISG and approved by the scientific management. The hMabs were selected for alkaline phosphatase (AP) expression and growth in Iscove's modified Dulbecco's medium (IMDM; Sigma) containing 10% FBS, 2 mM glutamine, 0.1 mM beta-mercaptoethanol, 1% nonessential

amino acids, human basic fibroblast growth factor (5 ng/ml), penicillin (100 IU/ml), streptomycin (100 mg/ml), 0.5 mM oleic and linoleic acids (Sigma), 1.5 mM $Fe^{2+}$ [iron(II) chloride tetrahydrate, Sigma; or Fer-In-Sol, Mead Johnson], 0.12 mM $Fe^{3+}$ [iron(III) nitrate nonahydrate, Sigma; or Ferlixit, Aventis] and 1% insulin/transferrin/selenium (Gibco).

### PEG-fibrinogen

PEG-fibrinogen precursor solution was prepared and photopolymerized as described elsewhere (Corona *et al*, 2012). We prepared PEG hydrogels containing Mabs by mixing a PBS cell suspension and PEGylated fibrinogen precursor solution containing 0.1% of Igracuret™2959 photoinitiator (Ciba Specialty Chemicals) to have a final concentration of 8 mg/ml with the desired cell concentration. We added 30 ml aliquots of the suspension into cylindrical silicon moulds and placed them under a long-wave UV lamp (365 nm, 4–5 mW/cm$^2$) for 5 min in a laminar flow hood. DMEM culture medium (containing 10% FBS) was added immediately to the polymerized hydrogels to ensure cell growth for the *in vitro* experiments. The plugs were cultured for 24 h in serum-supplemented growth medium and then transferred into serum-depleted differentiation medium for 5 days in order to promote muscle fibre formation; for *in vivo* experiments, the moulds were directly implanted in the animals subcutaneously on the back, underneath the skin on the surface of the *tibialis anterior* (TA) or in its anatomical lodge, after ablation, without *in vitro* culture.

### Surgical procedure

Two-month-old male RAG2/γchain (mouse Mabs) or SCID (human Mabs) transgenic mice were provided, respectively, by Taconic and Charles Rivers, they were bred in ventilated cage at the Plaisant SPF (Specific Pathogen Free) animal house of Castel Romano. Mice were anesthetized with an intramuscular injection of physiologic saline (10 ml/kg) containing ketamine (5 mg/ml) and xylazine (1 mg/ml) and then implanted subcutaneously on the back, or underneath the skin on the surface of the TA with PF constructs containing $1.5 \times 10^6$ Mabs-nLacZ/PlGF or hMabs. In order to ensure a good placement of the construct, we performed a limited incision on the medial side of the back or the leg, separated the dorsal muscle or the TA from the skin and placed the plug constructs as desired and finally sutured. For the artificial muscle atrophy and hypertrophy, RAG2/γchain null mice implanted with PF constructs containing $1.5 \times 10^6$ Mabs-nLacZ/PlGF were subjected to denervation procedure and clenbuterol (Sigma) administration, respectively, after 4 weeks of the artificial muscle development. The right sciatic nerve was isolated in the mid-thigh region and cut, leading to denervation of the lower limb muscles. Clenbuterol was administered at a dose of 2 mg/kg/day via intraperitoneal injection, continuously for 1 month. To evaluate the host contribution in artificial muscle development, PF constructs containing $1.5 \times 10^6$ mMabs were implanted on the surface of TA of ubiquitous GFP mice.

For TA removal, mice were anesthetized as previously described, limited incision on the medial side of the leg has been performed in order to reach the TA, and then utilizing a cautery to avoid bleeding, the muscle fibres were completely removed, PF constructs containing $3 \times 10^6$ mMabs-PlGF were polymerized in the TA venue as

already described and then the incision sutured. All the experimental animal groups are summarized in Supplementary Table S2. Analgesic treatment (Rimadyl, Pfizer, USA) was administered after the surgery to reduce pain and discomfort. Mice were sacrificed at different time points for molecular and morphological analysis. Experiments on animals were conducted according to the rules of good animal experimentation I.A.C.U.C. no 432 of 12 March 2006.

### Satellite cells isolation

For artificial muscle satellite cell isolation, 8-week injured artificial and TA muscles ($n = 3$) were minced and digested in HBSS (Gibco) containing 2 µg/ml collagenase A (Roche), 2.4 U/ml dispase I (Roche), 10 ng/ml DNase I (Roche), 0.4 mM CaCl$_2$ and 5 mM MgCl$_2$ for 1 h at 37°C. Muscle satellite cells were magnetically sorted as CD45$^-$, CD31$^-$, α7-integrin$^+$ cells. First, the CD45$^+$ cells were magnetically labelled with anti-CD45 and CD31 MicroBeads (Miltenyi). Then, the cell suspension was loaded onto a MACS column (Miltenyi), which is placed in the magnetic field of a MACS separator. The magnetically labelled CD45, CD31$^+$ cells were retained within the column. The unlabelled cells run through. Then, these cells were labelled with anti-α7-integrin MicroBeads (Miltenyi) and loaded onto a MACS column, which was placed in the magnetic field of a MACS separator. After removing the column from the magnetic field, the magnetically retained α7-integrin$^+$ cells have been eluted as the positively selected cell fraction.

### Histology and immunocytochemistry

Cells and tissues were fixed in PFA 2% and processed for histology and immunocytochemistry as previously described (Gargioli *et al*, 2008). The primary antibodies used were mouse anti-Pax7 (DSHB) at 1:20, mouse MF20 (DSHB) at 1:2, rabbit anti-laminin (SIGMA #9393) at 1:100, rabbit anti-LacZ (Cappel) at 1:100, rat anti-VE-cadherin (clone BV13 homemade) at 1:100, mouse anti-SMA (Sigma) at 1:100, mouse anti-dystrophin (Vector) at 1:100, mouse anti-neuronal class III β-tubulin (COVANCE) and alpha-bungarotoxin Alexa594 (Molecular probes) at 50 mg/ml. The secondary antibodies used at 1:100 were anti-mouse Alexa555 (Molecular Probes), anti-rabbit Alexa488 (Molecular Probes) and anti-rat Alexa568 (Molecular Probes), Cy2-anti-mouse (Jackson), AMCA-anti-mouse (Jackson), goat anti-mouse horseradish peroxidase (HRP)-conjugated IgG (Bio-Rad) for immunohistochemistry against MF20, developing the peroxidase reaction by AEC (3-amino-9-ethylcarbazole) substrate (SIGMA). The sections were photographed with Nikon ECLIPSE 2000-TE microscope or with Olympus FV 1000 confocal laser scanning microscope for the confocal images. VE-cad-positive capillary endothelial cells were counted under fluorescence microscopy (×200) in five randomly selected fields of different sections from each sample and related to the number of muscle fibres in the same section (Díaz-Manera *et al*, 2010).

### Immunoblotting

Tissue samples from artificial and control muscles were first triturated in liquid nitrogen and immediately homogenized with RIPA buffer (20 mM Tris–HCl, pH 7.4, 5 mM EDTA, 0.1% SDS, 1% NP-40, 1% NaDOC and protease inhibitor cocktail; Roche).

Homogenates and centrifuged at 12,000 $g$ for 10 min at 4°C to discard nuclei and cellular debris. Protein concentration was determined by bicinchoninic acid (BCA) protein assay (Pierce) using bovine serum albumin as standard. Total homogenates were separated by sodium dodecyl sulphate–polyacrylamide gel electrophoresis (SDS–PAGE) with a concentration opportunely chosen on the base of molecular weight of the proteins analysed. For Western blot analysis, proteins were transferred to Immobilon (Amersham) membranes, saturated with 5% non-fat dry milk (Bio-Rad), 0.1% Tween-20 (Sigma) in PBS (blocking solution) and hybridized with α-creatine kinase MM polyclonal antibody (Abcam ab83441), with tropomyosin 4 polyclonal antibody (Chemicon ab5449), with troponin I mouse monoclonal antibody (Chemicon MAB1691), with GAPDH (clone GAPDH-71.1; Sigma), with VE-cadherin rat monoclonal antibody (clone BV13 homemade) at 1:1,000 dilution or with MF20 mouse monoclonal antibody at 1:5 for 1 h at RT. The filters were washed three times (15 min each at RT) with wash solution (PBS 0.1% Tween-20) and then reacted with anti-mouse, anti-rat or anti-rabbit secondary antibody conjugated with HRP IgG (Bio-Rad) at 1:3,000 dilution for 1 h at RT, washed three times and finally visualized with the ECL. The MyHC isoform composition of the single muscle fibre segments used in mechanical experiments was determined with a procedure previously described (Carosio et al, 2013). In the MyHC region, four bands corresponding to the four adult MyHC isoforms (MyHC-1, MyHC-2A, MyHC-2X and MyHC-2B) could be separated. In relation to the presence of one or two bands in the MyHC region, fibres were classified into four pure fibre types and three hybrid fibre types: 1, 2A, 2X and 2B (or pure fibres), and 1-2A, 2AX and 2XB (or hybrid fibres). The same electrophoretic protocol followed by densitometry analysis of MyHC bands was used to determine the MyHC isoform composition of whole muscle samples as previously described (Harridge et al, 1996).

## Methods used to study force (Po), CSA and maximum shortening velocity (Vo) of single skinned mice fibres from artificial muscles and from controls (CTR)

The analysis of single muscle fibres was performed according to a procedure previously described (Hughes & Blau, 1990; Pellegrino et al, 2003). Briefly, after animal sacrifice, muscle bundles to be used for single fibre analysis were stored at −20°C in a solution containing skinning solution and glycerol (50%). On the day of the experiment, a bundle was transferred to a dish containing skinning solution, maintained at 10°C, and single muscle fibres were dissected with the help of a stereomicroscope (Wild M3 10–60× magnification). Exposure for 30 min to skinning solution containing Triton X-100 1% was used to ensure membrane solubilization. Segments ~1 mm of length were cut from the fibres, and light aluminium clips were applied at both ends of the segments to attach them to the beams of the force transducer (AE 801 SensoNor, Horten, Norway) and of the isotonic lever (model 101 vibrator; Ling Dynamic System, Royston, UK) in the experimental set-up. Skinning (5 mM EGTA, pCa 9.0), relaxing (5 mM EGTA, pCa 9.0), pre-activating (EGTA 0.5 mM, pCa 9.0) and activating (EGTA 5 mM pCa 4.5) solutions were prepared as previously described (Pellegrino et al, 2003). The experimental set-up enabled quick transfer of the muscle fibres from the first, larger chamber (~0.4 ml), containing skinning solution, to three, smaller chambers

(70 ml), containing relaxing, pre-activating and activating solution. The electromagnetic puller could either keep the length of the fibre segment constant to elicit isometric contractions, or impose to the specimen quick releases of preset amplitude completed in 2 ms. A stereomicroscope was fitted over the apparatus to view the fibre at 20–60× magnifications during the mounting procedure and during the experiment. The set-up was placed on the stage of an inverted microscope (Axiovert 10; Zeiss, Germany). As the floors of the muscle chambers were made by cover slips, specimens could be viewed at 320× magnification through the eyepieces of the microscope. The signals from the force and displacement transducer were fed into a personal computer using an AD converter (interface CED 1401 plus, Cambridge, UK), viewed through the computer screen and analysed by data analysis software (Spike 2, CED, Cambridge, UK). As skinned fibres lack plasma membrane, they need to be activated by exposure to solution containing the activating ion, $Ca^{2+}$. As intact fibres can hardly be dissected from skeletal muscles of small mammals, skinned fibres have been widely used to study contractile properties of muscle fibres from small mammals and humans and have proved to be very reliable specimen for this kind of analysis (Bottinelli et al, 1994; Pellegrino et al, 2003; Torrente et al, 2004). In mechanical experiments in the present work, temperature was set at 12°C. Sarcomere length (S.L.) was determined by counting striations in segments of known length at 320× magnification and set at 2.5 μm by varying fibre length at rest. CSA of the specimen was determined assuming a circular shape from the mean of the three diameters measured at 320× magnification, without correction for swelling. For force (Po) and maximum shortening velocity (Vo), determinations fibres were first transferred to pre-activating solution for at least 2 min and then maximally activated (pCa 4.45) for about 40–60 s. To determine Vo, slack-test manoeuvres were employed. Details of Vo determinations have been reported previously (Bottinelli et al, 1994; Torrente et al, 2004). Vo was expressed in fibre length per second (l/s), Po in mg and Po/CSA in kN/$m^2$. At the end of the mechanical experiment, fibres were put in 20 ml of standard buffer and stored at −20°C for subsequent analysis of MyHC isoform content using a procedure previously described (Bottinelli et al, 1994; Torrente et al, 2004).

## RT–PCR

The RNA was isolated from artificial muscles and tibialis anteriors of immunodeficient-treated mice. One microgram of RNA extracted with TRIzol (Invitrogen) was converted into double-stranded cDNA with the cDNA synthesis kit ImProm™-II Reverse Transcription System (Promega), according to the manufacturer's instructions, using quantitative real-time PCR. RNA was retrotranscribed as described above. Quantitative PCRs were performed with a real-time PCR thermocycler (Mx3000P; Stratagene). Each cDNA sample was amplified in triplicate using GoTaq qPCR master mix (Promega).
 Primers used are listed below:
 MyLC3F (113 bp): forward CCACTCAGGGATTGGAGCTGCCT reverse GCCTCCTTGAAGTCGGCAATCTGG
 MyHC (160 bp): forward GGCCAAAATCAAAGAGGTGA reverse CGTGCTTCTCCTTCTCAACC
 Dys (336 bp): forward TCTCATCGTACCTAAGCCTC CAGTGCCTTGTTGACATTGTTCAG
 Tpm alpha (173 bp): forward CGGGCTGAGCTCTCAGAAGGC

reverse GCCCGAGTTTCAGCCTCCTTCA
Tpma beta (271 bp): forward GAAAGAGGCTGAGACCCGAGCAGA
reverse TCAGGCTGGCTGTGCAACGTG
Gapdh (250 bp): forward TTCACCACCATGGAGAAGGC
reverse GGCATGGACTGTGGTCATGA
nLacZ (175 bp): forward ATCTCTATCGTGCGGTGGTT
reverse GAGCTGACCATGCAGAGGAT

**Regeneration model**

Mabs-PF implanted (8 weeks) RAG2/γchain null mice were injected with 20 ml of 10 mM cardiotoxin (Latoxan) in PBS, into the artificial muscle: limited incision in the medial side of the leg was performed in order to visualize artificial muscle and inject cardiotoxin into it. The treated artificial muscles were then collected 3, 7 and 14 days following cardiotoxin injection. The muscles were harvested and analysed to investigate artificial muscle fibres regeneration.

**Treadmill**

A 7° uphill treadmill protocol was performed using an Exer-3/6 open treadmill (Columbus Instruments, Columbus, OH, USA) according to guidelines from the American Physiological Society. The treadmill protocol consisted of five continuous days of incremental training followed by experimental determination of maximal running distance on day 6. Mice were first run at 10 m/min for 20 min, and treadmill speed was then increased by 1 m/min every 2 min until mice were exhausted. Exhaustion was defined as spending > 10 s on the shocker without attempting to re-enter the treadmill (McCullagh *et al*, 2008; Zeng *et al*, 2014).

**Fore-hind leg grip test**

A computerized grip-strength meter (Columbus Instruments) was used to measure fore-hind limb grip strength in conscious mice. Mice were acclimatized for 5 min before starting test. Gently the mouse was lowered over the top of the grid so that both its front paws and hind paws are allowed to grip the smooth metal pull bar at the top of the apparatus. The mouse was then gently pulled backward in the horizontal plane until it could no long grasp the bar. The force at the time of release was recorded as the peak tension. Each mouse was tested five times with a 20–40 s break between tests. The average peak tension from three best attempts was defined as fore-hind limb grip strength.

**Cross-sectional area analysis**

Morphometrical analyses to evaluate means of fibre per CSA were carried out on 800 fibres per muscle on laminin-stained sections of artificial muscle or TA using ImageJ (NIH) software. The CSA means were scored in three non-adjacent transverse sections from the largest muscle portion for three mice per experimental group.

**Statistical analysis**

Data were analysed using GraphPad Prism 5, and values were expressed as means ± standard error (SEM). Statistical significance was tested using Student's *t*-test. A probability of less than 5%

**The paper explained**

**Problem**
Different causes such as major traumatisms or surgery for cancer result in extensive loss of skeletal muscle tissue that cannot be repaired by the muscle itself. This unmet clinical need has stimulated many attempts to re-create a functional muscle either outside or directly inside the body. This has so far been a significant challenge because it is difficult to reconstruct the complex architecture of the skeletal muscle tissue that has a high requirement of oxygen.

**Results**
We combined muscle progenitors (mesoangioblasts), engineered to stimulate blood vessel growth from the host, with a hydrogel (PEG-fibrinogen). When the graft was implanted underneath the skin on the surface of a normal, contracting skeletal muscle (*tibialis anterior*), mature and aligned muscle fibres formed within several weeks as a complete and functional extra muscle. Moreover, replacing the *tibialis anterior* ablated with the same combinations of cells and hydrogel, we obtained a complete and functional recovery of the ablated muscle.

**Impact**
This strategy opens the possibility for patient-specific muscle generation for a large number of pathological conditions involving muscle tissue wasting. It should however be considered that a mouse muscle is very small and scaling up the process may require significant additional work.

($P < 0.05$) was considered to be statistically significant. All the experimental mice were analysed, and inclusion/exclusion criteria were based on the presence or absence of artificial muscle generation after implantation procedure. Blind analyses were conducted for quantification of PlGF effect on vessel density, clenbuterol treatment and denervation procedure affecting muscle CSA in order to minimize investigator bias.

**Supplementary information** for this article is available online: http://embomolmed.embopress.org

**Acknowledgements**
We thank M. Coletta for technical assistance, E. Dejana for the gift of BV13 antibody, the centre of Advanced Microscopy 'P. Albertano' for confocal imaging, M.C. Panzeri and Alembic for electron microscopy imaging, R. Biagini and C. Zoccali for providing human biopsies and surgical advices and L. Madaro for CSA tips. This work was supported by EC-IP FP7 grants Biodesign (to DS and GC), Duchenne Parent project Italia to GC and by the Cariplo Foundation, Italy (grant no. 2010.0764) and by the European Commission, MYOAGE grant (no. 223576), funded under FP7 to RB.

**Author contributions**
GC and CG designed the research and wrote the paper. CF prepared vectors, transduced cells and carried out most of the experimental work; RR and CB performed artificial muscle functional analysis and helped with data interpretation; AB tested different PF hydrogel concentration; EL, AM and RB designed and performed physiological experiments; KS-S, OK and DS produced and implemented PF for muscle experiments, tested *in vitro* PF compositions and performed confocal time-lapse experiments; CL, MLS, SS and ST did the histology and staining; SB did cell culture and immunostaining; CG and RR performed surgical operation; SMC and DS helped with study design, data analysis interpretation and paper writing.

## Conflict of interest

The authors declare that they have no conflict of interest.

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
