## [Review Process File · EMBO Molecular Medicine]

In vivo generation of a mature and functional artificial skeletal muscle.

Claudia Fuoco, Roberto Rizzi, Antonella Biondo, Emanuela Longa, Anna Mascaro, Keren Shapira-Schweitzer, Olga Kossov, Sara Benedetti, M. Lavinia Salvatori, Sabrina Santoleri, Stefano Testa, Sergio Bernardini, Roberto Bottinelli, Claudia Bearzi, Stefano M. Cannata, Dror Seliktar, Giulio Cossu and Cesare Gargioli

Corresponding authors: Giulio Cossu, University of Manchester; Stefano M Cannata, Tor Vergata Rome University; and Cesare Gargioli, Tor Vergata Rome University

Review timeline:

Submission date:	13 March 2014
Editorial Decision:	24 April 2014
Revision received:	12 December 2014
Editorial Decision:	12 January 2015
Revision received:	16 January 2015
Accepted:	22 January 2015

Transaction Report:

Editor: Céline Carret

1st Editorial Decision

24 April 2014

Thank you for the submission of your manuscript to EMBO Molecular Medicine. We finally have heard back from the three referees whom we asked to evaluate your manuscript. Although the referees find the study to be of interest, they also raise a number of concerns that will have to be satisfactorily addressed in the next version of your manuscript.

As you will from the enclosed reports, while referee 1 is supportive, together with referee 3, they would like you to perform additional experiments to clarify the stem cells fate. In addition, referee 1 suggests demonstrating the physiological response of implanted muscles in response to different stimuli, a concern partially shared by referee 2. Referee 2 is also concerned about the advance of the findings and we would suggest rephrasing the main text and adding the appropriate references as suggested. Referee 3 also requires more details and better explanations here and there. Furthermore, this last referee suggests addressing the proof-of-principle of the findings as suggested in point 3. However, the main important issue of the current manuscript resides in the unclear clinical relevance as highlighted by both referees 2 and 3, although for different reasons. As such, I would strongly encourage you to address this fact to the best of your abilities as this is critical to our scope.

Given these evaluations, I would like to give you the opportunity to revise your manuscript, with the understanding that the referees' concerns must be fully addressed and that acceptance of the manuscript would entail a second round of review. However, please note that that it is our journal's

policy to allow only a single round of revision, and that acceptance or rejection of the manuscript will therefore depend on the completeness of your response and the satisfaction of the referees with it.

I look forward to seeing a revised form of your manuscript as soon as possible.

***** Reviewer's comments *****

Referee #1 (Comments on Novelty/Model System):

In this manuscript, Fuoco et al. present interesting evidence on formation of "artificial" skeletal muscle from PEG-fibronogen hydrogenl-embedded mesoangioblasts. This muscle shows several properties of "natural" muscle, both in vitro and when implanted in recipient mice, where they appear to contribute to regeneration and to functionally replace fully ablated muscles. This is to my knowledge the first evidence of skeletal muscle repair with a bioengineering strategy. As such, the manuscript is of clear interest to the bio-medical audience.

Referee #1 (Remarks):

In this manuscript, Fuoco et al. present interesting evidence on formation of "artificial" skeletal muscle from PEG-fibronogen hydrogenl-embedded mesoangioblasts. This muscle shows several properties of "natural" muscle, both in vitro and when implanted in recipient mice, where they appear to contribute to regeneration and to functionally replace fully ablated muscles. This is to my knowledge the first evidence of skeletal muscle repair with a bioengineering strategy. As such, the manuscript is of clear interest to the bio-medical audience.

I encourage the authors to revise this manuscript by addressing the following points.

- 1) The finding that an "artificial" muscle generates satellite cells that confer regeneration ability is of special importance and needs to be supported by further data. It implies that a fraction of mesoangioblasts can adopt the "satellite cell fate", and this is intriguing, but deserves more careful investigation. The authors need to isolate the Pax7 positive cells from implanted muscles after injury of the implanted muscles itself, and possibly also after injury of the underlying endogenous muscles, assuming that regeneration cues should be "sensed" by the near implanted muscle. Then, a standard characterization of satellite cell potential should be carried.
- 2) To conclusively demonstrate that the implanted muscles behave as an endogenous muscle, the authors should provide evidence that it undergoes physiological changes of size that reflect their response to systemic factors. For instance, it would be important to show that once implanted, the artificial muscle undergo hypertrophy or atrophy, in response to systemic stimuli (experimental anabolic and catabolic conditions) to an extent comparable to the endogenous muscles.
- 3) Figures are not numbered, please include numbers
- 4) In figure 3 (I assume this is the number), panel J, lane 3 is clearly introduced by photoshop artwork. This could be fine, as long as the authors disclose it and state from what experiments lanes 1 and 2 and lane 3 have been derived.

Referee #2 (Remarks):

This is a potentially interesting manuscript that describes a large series of experiments conducted using mesangioblasts in combination with a PEG-fibrinogen hydrogel scaffold to mediate muscle tissue regeneration in a mouse TA injury model. The *in vitro* work is well done and the histological and molecular features of the observed regeneration are well described and illustrated. However, as intriguing as the technology may be, the importance of the findings to the field, as well as their potential clinical implications are grossly overstated. My specific comments are below:

Major Comments:

1. There is now a fairly large and substantive literature regarding experimental strategies for treatment of traumatic muscle injury. The authors do not acknowledge, nor cite, the vast majority of prior work in the field. In fact, diverse scaffolds/biomaterials, as well as a variety of cell types- both administered alone and in various combinations, have already been explored in a variety of small and large animal models with varying levels of functional and tissue recovery reported. The fact that there is such a large and diverse literature (see Shireman, Walters, Blau, Bursac, Corona, de Coppi, Christ, Badylak, Merritt, Dennis, among others) clearly indicates that the current work is not the "ground-breaking advance over previous attempts..." the authors claim. The nature of their findings must be placed in proper context of the other developments in the field.
2. The authors also note that their work: "...shows the unprecedented possibility to repair a massive muscle ablation with a functional *resitutio ad integrum*". The precise size of the defect created was not stated anywhere- investigators will generally report the amount of tissue removed to provide an indication of the size of the defect created. From what is described in the text, the implant was at most 10 mm², and based on prior work in the mouse model the actual amount of tissue removed from the TA was likely in the neighborhood of 4 mg. It is not clear how this can be considered a "massive" ablation, nor is there any human clinical condition that would be relevant to the studies described herein. In fact, these injuries are so small that absence of specific strategies for vascularization and innervation are likely not even that limiting- given sufficient time to heal. The entire premise of the significance of this technology to the treatment of traumatic injuries, while certainly interesting, is greatly exaggerated based on the data presented.
3. While the authors are to be commended for attempting to establish the physiological significance of their technology there are still some issues remaining. First, with respect to the single fiber studies, although the number of experiments is incredibly impressive, no literature reference or guidance is provided as to what the expected specific force should be for single myofibers from the TA. As the authors may be well aware, the values for specific force can vary substantially among different muscle types and species. What is the native single myofiber specific force expected to be for the mouse TA muscle? Is there any reference point at all? If such information is not available for this muscle then is there a known value for a comparable muscle? Second, Given that the TA is not a weight bearing muscle for treadmill training what is the evidence that treadmill training has anything to do with TA function? The same is true for the grip test- the TA is responsible for dorsiflexion of the foot, so what is role of the TA muscle in grip? What is the rationale for believing that the grip test will provide useful information about the functional recovery of the TA muscle? A standard test used to study TA muscle functional recovery following traumatic injury or other impairments to the muscle is *in vivo* dorsiflexion of the foot following peroneal nerve stimulation. An additional advantage of this quantitative measure is that it is also an excellent index of functional re-innervation. Finally, there is no discussion concerning the extent to which synergistic muscles (EDL and EHL) might contribute to any expected functional recovery.

Minor concerns:

1. A schematic diagram depicting the process, time lines and sequence of events associated with creation of the PEG-PF constructs and addition of cells (prior to implantation) would be very helpful.
2. A table summarizing all of the experimental groups and time points, number of animals per group, and methods of study would also be most useful.

3. Many of the histological and immunochemical figures had no scale bar on them. I do not know if that is an artifact of the PDF creation process, but it made it very difficult to appreciate the findings.
4. Similarly, in several cases there were no indices provided for statistical significance on bar graphs, etc. for comparisons of interest.
5. The units for the physiological testing in Fig. 4 are not shown for the Y-axis. (at least on the graphs I have in front of me). The symbol for the WT animals is not shown for the treadmill testing.

Referee #3 (Comments on Novelty/Model System):

please see general remarks

Referee #3 (Remarks):

Using the body as a 'bioreactor' the authors report a tissue engineering strategy for the de novo muscle tissue formation in order to repair or strengthen damaged skeletal muscle. Muscle progenitor cells were embedded in a hydrogel matrix ('PEG-fibrinogen') which was placed on the surface of existing muscle tissue (or under the skin). The in vivo microenvironment promoted the conversion of this cell/biomaterial construct into 'artificial' functional muscle tissue which was shown to have similar characteristics than native tissue. The strategy seemed to work well in the case of mouse cells and, to a lesser extent, human cells.

Overall, the manuscript is well written and of interest to the readership of EMBO Molecular Medicine. I recommend publication of this manuscript, provided that the following concerns can be addressed:

1. The in vitro behavior of mesoangioblasts encapsulated in PEG-fibrinogen is poorly described. The authors briefly mention that adult mouse mesoangioblasts "showed good survival and muscle differentiation" and show two images (Supplementary Fig. 1A and 1B) that should support this claim. Firstly, it is not clear what these images really show as the cells look like they were cultured on the surface rather than within a 3D scaffold. Secondly, since PEG-fibrinogen is apparently crosslinked via UV light, the reader is left to wonder whether viability is really not an issue in this system. The authors should provide quantitative data on cell viability after encapsulation as well as on in vitro differentiation and how these read-outs compare to conventional 2D cultures.
2. The authors claim that differentiation occurs much faster in 3D scaffolds compared to 2D which is very surprising, given that the gels appear to be very dense compared to previous tissue engineering strategies which were based on porous collagen or matrigel scaffolds. However, the only data supporting this claim are supplementary videos that do not show much. The authors should provide quantitative data differentiation dynamics in 2D and 3D. Also, it would be very interesting to the readership to understand how cells are growing so efficiently within these hydrogel scaffolds. What happens to the gels over time in culture? Are they degraded and replaced by cell-secreted proteins? How are they degraded?
3. One aspect that significantly diminishes my enthusiasm for this approach is that it seems to be dependent on cells engineered to express Placenta derived Growth Factor. This limits the clinical applicability of the approach quite substantially. First, can artificial muscle also be obtained in vivo from wild-type mesoangioblasts? Second, could recombinant Placenta derived Growth Factor be delivered from PEG-fibrinogen containing mesoangioblasts in order to bypass the engineering of the cells? Proof-of-principle experiments addressing these questions would significantly increase the impact of this study.
4. Due to incomplete transduction of the mMabs-nLacZ used in the in vivo studies, it is unclear to which extent the artificial muscle was obtained by migration of host-derived cells. This question could be readily addressed by labeling host cells and quantifying their contribution to the artificial muscle tissue.

ITEMIZED REPLIES TO REFEREES COMMENTS (IN BLUE)

Referee #1 (Comments on Novelty/Model System):

In this manuscript, Fuoco et al. present interesting evidence on formation of "artificial" skeletal muscle from PEG-fibronogen hydrogenl-embedded mesoangioblasts. This muscle shows several properties of "natural" muscle, both in vitro and when implanted in recipient mice, where they appear to contribute to regeneration and to functionally replace fully ablated muscles.

This is to my knowledge the first evidence of skeletal muscle repair with a bioengineering strategy. As such, the manuscript is of clear interest to the bio-medical audience.

Referee #1 (Remarks):

In this manuscript, Fuoco et al. present interesting evidence on formation of "artificial" skeletal muscle from PEG-fibronogen hydrogenl-embedded mesoangioblasts. This muscle shows several properties of "natural" muscle, both in vitro and when implanted in recipient mice, where they appear to contribute to regeneration and to functionally replace fully ablated muscles.

This is to my knowledge the first evidence of skeletal muscle repair with a bioengineering strategy. As such, the manuscript is of clear interest to the bio-medical audience.

I encourage the authors to revise this manuscript by addressing the following points.

1) The finding that an "artificial" muscle generates satellite cells that confer regeneration ability is of special importance and needs to be supported by further data. It implies that a fraction of mesoangioblasts can adopt the "satellite cell fate", and this is intriguing, but deserves more careful investigation. The authors need to isolate the Pax7 positive cells from implanted muscles after injury of the implanted muscles itself, and possibly also after injury of the underlying endogenous muscles, assuming that regeneration cues should be "sensed" by the near implanted muscle. Then, a standard characterization of satellite cell potential should be carried.

Following Referee suggestion, we isolated satellite cells from artificial and underlying TA muscle after CTX induced damage. However, because we do not have available the Pax7-GFP mouse, we cultured mononucleated cells from injured artificial and underlying TA both at high and clonal density as described (Cossu et al. Dev. Biol. 1989). As shown in the supplementary figure 9 we now demonstrate Pax7 expression in proliferating satellite cells and MyHC expression differentiated myotubes in both host TA-derived (LacZ-) and in artificial muscle-derived (LacZ+) cells.

2) To conclusively demonstrate that the implanted muscles behave as an endogenous muscle, the authors should provide evidence that it undergoes physiological changes of size that reflect their response to systemic factors. For instance, it would be important to show that once implanted, the artificial muscle undergo hypertrophy or atrophy, in response to systemic stimuli (experimental anabolic and catabolic conditions) to an extent comparable to the endogenous muscles.

Following reviewer advice we performed hypertrophy and atrophy inducing experiments. The results, supporting the hypothesis, are shown in supplementary figure 10.

3) Figures are not numbered, please include numbers

Numbers have been included.

4) In figure 3 (I assume this is the number), panel J, lane 3 is clearly introduced by photoshop artwork. This could be fine, as long as the authors disclose it and state from what experiments lanes 1 and 2 and lane 3 have been derived.

Respectfully, figure 3 J is not a Photoshop collage, but the same gel that is now enclosed in its original issue during SDS PAGE procedure:

Referee #2 (Remarks):

This is a potentially interesting manuscript that describes a large series of experiments conducted using mesangioblasts in combination with a PEG-fibrinogen hydrogel scaffold to mediate muscle tissue regeneration in a mouse TA injury model. The *in vitro* work is well done and the histological and molecular features of the observed regeneration are well described and illustrated. However, as intriguing as the technology may be, the importance of the findings to the field, as well as their potential clinical implications are grossly overstated. My specific comments are below:

Major Comments:

1. There is now a fairly large and substantive literature regarding experimental strategies for treatment of traumatic muscle injury. The authors do not acknowledge, nor cite, the vast majority of prior work in the field. In fact, diverse scaffolds/biomaterials, as well as a variety of cell types- both administered alone and in various combinations, have already been explored in a variety of small and large animal models with varying levels of functional and tissue recovery reported. The fact that there is such a large and diverse literature (see Shireman, Walters, Blau, Bursac, Corona, de Coppi, Christ, Badylak, Merritt, Dennis, among others) clearly indicates that the current work is not the "ground-breaking advance over previous attempts..." the authors claim. The nature of their findings must be placed in proper context of the other developments in the field.

Respectfully, we do not entirely agree with the Reviewer. It is true that an abundant literature on the topic exists and we apologize for not having quoted all the relative papers. This has now been amended. We have also eliminated the term "ground breaking". That said, the Reviewer should agree with us that what we obtain is a significant step forward in the field. It is not sufficient to see a couple of muscle fibers to claim that an artificial muscle has been generated.

Moreover, in the context of massive muscle loss we do really believe that our work may have a relatively rapid impact in clinical work for muscle regeneration due to separate clinical trials performed with PEG-Fibrinogen and human mesoangioblast respectively for cartilage and muscle defects recovery. Finally, Badylak lab (Sicari et al., 2014) published a very recent paper on mice and human treatment for volumetric muscle loss (VML) damage recovery using a tissue engineering approach. This has been quoted.

2. The authors also note that their work: "...shows the unprecedented possibility to repair a massive

muscle ablation with a functional *resitutio ad integrum*". The precise size of the defect created was not stated anywhere- investigators will generally report the amount of tissue removed to provide an indication of the size of the defect created. From what is described in the text, the implant was at most 10 mm² (sicari... badylak 2012 3x4mm damage VML), and based on prior work in the mouse model the actual amount of tissue removed from the TA was likely in the neighborhood of 4 mg. It is not clear how this can be considered a "massive" ablation, nor is there any human clinical condition that would be relevant to the studies described herein. In fact, these injuries are so small that absence of specific strategies for vascularization and innervation are likely not even that limiting- given sufficient time to heal. The entire premise of the significance of this technology to the treatment of traumatic injuries, while certainly interesting, is greatly exaggerated based on the data presented.

The term "massive" clearly refers to the mouse (and this is now clarified in the text). In the specific case, the anterior TA normally weights around 100mg and the removed TA muscle tissue weighted around 80-90mg so that the defect created was approximately 80-90% of the entire TA as shown in supplementary figure 12. Since the control implanted with only PEG-Fibrinogen showed almost complete absence of any muscle tissue replacing the dislodged TA while that with PEG-Fibrinogen and Mesoangioblasts has the same size of the removed TA, we may conclude that the significance of this technology is not exaggerated but, once scaled up, a real promise for tissue engineering of human muscles, though admittedly, for only small muscles at least for the next future.

3. While the authors are to be commended for attempting to establish the physiological significance of their technology there are still some issues remaining. First, with respect to the single fiber studies, although the number of experiments is incredibly impressive, no literature reference or guidance is provided as to what the expected specific force should be for single myofibers from the TA. As the authors may be well aware, the values for specific force can vary substantially among different muscle types and species. What is the native single myofiber specific force expected to be for the mouse TA muscle? Is there any reference point at all? If such information is not available for this muscle then is there a known value for a comparable muscle? Second, Given that the TA is not a weight bearing muscle for treadmill training what is the evidence that treadmill training has anything to do with TA function? The same is true for the grip test- the TA is responsible for dorsiflexion of the foot, so what is role of the TA muscle in grip? What is the rationale for believing that the grip test will provide useful information about the functional recovery of the TA muscle? A standard test used to study TA muscle functional recovery following traumatic injury or other impairments to the muscle is *in vivo* dorsiflexion of the foot following peroneal nerve stimulation. An additional advantage of this quantitative measure is that it is also an excellent index of functional re-innervation. Finally, there is no discussion concerning the extent to which synergistic muscles (EDL and EHL) might contribute to any expected functional recovery.

We agree that the values of specific force of individual muscle fibres can somewhat vary among different muscle types and species. They also vary among papers due to different experimental conditions, i.e temperature, and bathing solutions used. However, the observation that muscle fibres from artificial muscles and from the underlying TA muscles develop very similar specific force and have very similar maximum shortening velocity strongly suggest that artificial muscle fibres are functionally indistinguishable from normal muscle fibres. Moreover, the values of specific force reported here for both artificial and normal muscle fibres, ~ 60 kN/m², are fully consistent with values we previously reported for individual muscle fibres dissected from TA muscles of adult mice (Torrente et al. (2004). Human circulating AC133(+) stem cells restore dystrophin expression and ameliorate function in dystrophic skeletal muscle. *J Clin Invest* 114, 182-195). Finally, in the experimental conditions used in the present study, we previously reported specific force values very close to 60 kN/m² for muscle fibres of different species and muscles: fast human muscle fibres (61 kN/m²; Bottinelli R, et al. (1996). Force-velocity properties of human skeletal muscle fibres: myosin heavy chain isoform and temperature dependence. *J Physiol* 495 (Pt 2), 573-586); fast rat fibres (67.8 kN/m²; Reggiani C, et al. (1997). *J Physiol* 502 (Pt 2), 449-460); fast dog fibres (67 nK/m² Sampaolesi M et al (2006). Mesoangioblast stem cells ameliorate muscle function in dystrophic dogs. *Nature* 444, 574-579). We have added just a short comment and a reference in the text on this issue, but we can elaborate more if requested.

We inserted into material and methods references to the Treadmill as indicator of TA health and strength. Finally, since the TA is responsible for foot dorsiflexion of the foot, the movie enclosed

clearly show inability to perform this movement in TA-ablated animals. We enclose the movie (Supplementary Video 4). This also explains the reduced force on the grip test executed on the four limbs.

Minor concerns:

1. A schematic diagram depicting the process, time lines and sequence of events associated with creation of the PEG-PF constructs and addition of cells (prior to implantation) would be very helpful.

As suggested, a diagram has been added as supplementary fig. 3 and 13.

2. A table summarizing all of the experimental groups and time points, number of animals per group, and methods of study would also be most useful.

A table has also been added in supplementary materials (Supplementary table 1).

3. Many of the histological and immunochemical figures had no scale bar on them. I do not know if that is an artifact of the PDF creation process, but it made it very difficult to appreciate the findings

Respectfully, we would like to underline that in the original pictures there are scale bars with values expressed in the relative figure legends.

4. Similarly, in several cases there were no indices provided for statistical significance on bar graphs, etc. for comparisons of interest.

According to Referee criticism we modified images and indicate statistical significance in quantifying graph.

5. The units for the physiological testing in Fig. 4 are not shown for the Y-axis. (at least on the graphs I have in front of me). The symbol for the WT animals is not shown for the treadmill testing.

Regarding grip test there is newton (N) as units, while for treadmill testing there are not units neither WT symbol because the X axis represent the WT behavior and all the experimental point are expressed relative to WT exhaustion time and distance run.

Referee #3 (Comments on Novelty/Model System):

please see general remarks

Referee #3 (Remarks):

Using the body as a 'bioreactor' the authors report a tissue engineering strategy for the de novo muscle tissue formation in order to repair or strengthen damaged skeletal muscle. Muscle progenitor cells were embedded in a hydrogel matrix ('PEG-fibrinogen') which was placed on the surface of existing muscle tissue (or under the skin). The in vivo microenvironment promoted the conversion of this cell/biomaterial construct into 'artificial' functional muscle tissue which was shown to have similar characteristics than native tissue. The strategy seemed to work well in the case of mouse cells and, to a lesser extent, human cells. Overall, the manuscript is well written and of interest to the readership of EMBO Molecular Medicine. I recommend publication of this manuscript, provided that the following concerns can be addressed:

1. The in vitro behavior of mesoangioblasts encapsulated in PEG-fibrinogen is poorly described.

The authors briefly mention that adult mouse mesoangioblasts "showed good survival and muscle differentiation" and show two images (Supplementary Fig. 1A and 1B) that should support this claim. Firstly, it is not clear what these images really show as the cells look like they were cultured on the surface rather than within a 3D scaffold. Secondly, since PEG-fibrinogen is apparently crosslinked via UV light, the reader is left to wonder whether viability is really not an issue in this system. The authors should provide quantitative data on cell viability after encapsulation as well as on *in vitro* differentiation and how these read-outs compare to conventional 2D cultures.

We thank the Referee for pointing out the mistake in the text, i.e. the Mabs encapsulated into PF are shown in Fig. 1 rather than supplementary Fig. 1 that rather shows Mabs on standard plastic culture plastic. Indeed, in 2012 (Fuoco et al.) we published the results of the behavior of Mabs and other myogenic cells embedded into PF. Nevertheless, we performed time course analysis for Mabs growing in 2D standard plastic culture versus 3D PF embedded Mabs (Supplementary figure 1) showing quick myogenic differentiation into PF three-dimensional environment.

2. The authors claim that differentiation occurs much faster in 3D scaffolds compared to 2D which is very surprising, given that the gels appear to be very dense compared to previous tissue engineering strategies which were based on porous collagen or matrigel scaffolds. However, the only data supporting this claim are supplementary videos that do not show much. The authors should provide quantitative data differentiation dynamics in 2D and 3D. Also, it would be very interesting to the readership to understand how cells are growing so efficiently within these hydrogel scaffolds. What happens to the gels over time in culture? Are they degraded and replaced by cell-secreted proteins? How are they degraded?

As described above we performed time course on differentiating Mabs showing the effect of PF hydrogel on myogenic differentiation. PF is not very dense, it is a hydrogel rich in water and as published in Fuoco et al., 2012 we demonstrated that the material offers a suitable 3D environment promoting muscle differentiation in Mabs and other myogenic cells. Furthermore, we cultured PF *in vitro* up to 30 days: this revealed a slow and progressive degradation where the biomaterial was progressively replaced by cell-derived ECM. In contrast, *in vivo*, as we demonstrate here, after one week the PF was completely resorbed, likely due to the infiltration by phagocytic cells.

3. One aspect that significantly diminishes my enthusiasm for this approach is that it seems to be dependent on cells engineered to express Placenta derived Growth Factor. This limits the clinical applicability of the approach quite substantially. First, can artificial muscle also be obtained *in vivo* from wild-type mesoangioblasts? Second, could recombinant Placenta derived Growth Factor be delivered from PEG-fibrinogen containing mesoangioblasts in order to bypass the engineering of the cells? Proof-of-principle experiments addressing these questions would significantly increase the impact of this study.

At the time of writing there are 38 FDA approved clinical trials using lentivectors, (<http://www.clinicaltrials.gov/ct2/results?term=LENTIVIRAL+VECTORS&pg=1>), either running or completed. No SAE have been so far reported. By the time this strategy may reach clinical experimentation, lentivectors for cell genetic correction or modification will have become a routine reagent. Indeed, as we showed previously (Gargioli et al. Nature Med. 2008) PIGF is not essential but it helps survival of mesoangioblasts by stimulating angiogenesis.

4. Due to incomplete transduction of the mMabs-nLacZ used in the *in vivo* studies, it is unclear to which extent the artificial muscle was obtained by migration of host-derived cells. This question could be readily addressed by labeling host cells and quantifying their contribution to the artificial muscle tissue.

In order to address the Referee criticism, we implant unlabeled Mabs expressing PIGF into ubiquitous expressing GFP host mice. Due to the non immune-deficient nature of the host mouse, implants were rejected after less than a month (even though both mouse strain were C57Bl6). The experiment, even if for a short time, revealed the exclusive myogenic differentiation of implanted cells (Supplementary figure 7).

Nevertheless we can not exclude that at later time point satellite cells, fibroblasts or other host mononucleated cells might colonize the artificial tissue, but this should not be a problem and it might be less likely to occur in large animals or patients.

2nd Editorial Decision

12 January 2015

Thank you for the submission of your revised manuscript to EMBO Molecular Medicine. We sincerely apologise for the extreme delay in getting back to you. It happens rarely, but unfortunately no initial referee were able to re-review your manuscript despite multiple requests. This, added to the Holiday season in between, explain the delay. As I did not want to include yet another reviewer at this stage I asked for an expert advisor to help me reach a decision. This advisor now got back to me and agreed that your article is ready to be accepted pending editorial final amendments.

Please submit your revised manuscript within two weeks.

I look forward to reading a new revised version of your manuscript as soon as possible.